

# A fully Automated Dobson Sun Spectrophotometer for total column ozone and Umkehr measurements

René Stübi[1], Herbert Schill[2], Jörg Klausen[1], Eliane Maillard Barras[1], and Alexander Haefele[1]

[1]Federal Office of Meteorology and Climatology, MeteoSwiss, 1530 Payerne, Switzerland
[2]Physikalisch-Meteorologisches Observatorium / World Radiation Center, 7260 Davos Dorf, Switzerland

**Correspondence:** R. Stübi (rene.stubi@meteoswiss.ch)

**Abstract.** The longest ozone column measurements series are based on the Dobson sun spectrophotometers developed in the 1920s by Prof. G. B. W. Dobson. These ingenious and robustly designed instruments still constitute an important part of the global network presently. However, the Dobson sun spectrophotometer needs manual operation which leads to the discontinuation of its use at many stations. To overcome this problem, MeteoSwiss developed a fully automated version of the Dobson

5    spectrophotometer. The description of the data acquisition and automated control of the instrument is presented here with some technical details. The results of different tests performed regularly to control the instrument good working operation are illustrated and discussed.

Compared to manual operation, the automation results in higher frequency measurements with lower random error and additional housekeeping information to characterise the measuring conditions. The automated Dobson instrument allows a contin-

10   uous observation of the ozone column with a resolution of ∼1 DU unit under clear sky conditions


## 1 Introduction

Following the discovery of the thinning of the ozone layer and the implementation of the Montreal Protocol, the signatory countries committed themselves to continue monitoring the ozone layer state. Based on international cooperation, the global survey with multiple instruments on board satellites constitutes the main information source. However, the long development

period to prepare a satellite mission, its relatively short life time and the risk of failure once in orbit call for reference ground based measurements. Developed over the last 50 years and in response to this requirement, dedicated Dobson and Brewer networks are operated world wide to monitor the atmospheric state (*Dobson*, 1968; *Kerr et al.*, 1981; *Fioletov et al.*, 2005). Model simulations predict a recovery of the ozone layer that will lasts for several years or decades depending on the location on earth and on the altitude considered (*SPARC/IO3C/GAW*, 2019; *Pawson et al.*, 2014; *WMO*, 2018). Moreover, the uncertainties

associated to the climate change feedback on the ozone recovery process push back the life span of dedicated ground-based measurement networks for a sustained monitoring.

Most ozone-monitoring instruments are based on the absorption of the sun radiation in the UV part of the spectrum. The ozone layer located between 20 to 30 km altitude almost completely absorbs the radiation intensity below $\lambda \sim 300$ nm. Therefore, the ozone column largely controls the UV radiation intensity at the ground level. The principle of the instrument developed

by G. M. B. Dobson in the early 1920s is based on the measurements of the intensity of attenuated radiation at a number of narrow spectral bands. This was first recorded in with photographic plate in the earlier days, later on with photoelectric detectors and nowadays with photo-multipliers (PM) detectors. Many of the Dobson instruments manufactured in the second part of the 20th century are still used operationally and, together with Brewer instruments, constitute the backbone of the global ozone column monitoring network (*Komyhr*, 1980; *Komhyr et al.*, 1989).

The Dobson and Brewer instruments have the status of reference instruments to calibrate satellites, newer instruments or to adjust numerical models. Therefore the processing algorithm of the data is still subject to different analyses and improvements in the scientific literature. These mainly concern updates of the underlying ozone cross-sections, stray light bias corrections or the calibration process assuring the network homogeneity (*Redondas et al.*, 2014; *Moeini et al.*, 2019; *Christodoulakis et al.*, 2015).

Historically, the Arosa "Licht-Klimatisches Observatorium" (LKO) development was strongly linked to the Dobson instrument network extension promoted by G. M. B. Dobson in the first half of the 20th century (*Staehelin and Viatte*, 2019; *Brönnimann et al.*, 2003; *Perl and Dütsch*, 1958; *Scarnato et al.*, 2010). Dobson instruments have been operated since 1926 at Arosa, resulting in the longest continuous series worldwide. The primary objective of monitoring the ozone in the atmosphere was regularly questioned and the continuation of the measurements was jeopardized due to budget problems and evolving sci-

entific interests. Therefore, the measurements were also used for other purposes like weather forecasting or to understand the effect of sun radiation exposure to treat tuberculosis (*Staehelin et al.* , 2018). In the second half of the 20th century, the ozone layer thinning and climate change were the main justification for what is a major contribution of the LKO to the worldwide effort following the Montreal Protocol signature in 1988 (*Albrecht and Parker*, 2019; *Solomon*, 1999).





The more recent automation of the Dobson operation helped strengthening the support for the continuation of this activity. The expectations were a possible improvement of the data quality - mainly due to the increase of the measurements frequency and the independence from operator influence - as well as reduced operation costs.

The present publication describes the successful MeteoSwiss developments of an automated version of the Dobson sun
5 spectrophotometer. Today, the three instruments from LKO are fully automated for the ozone column measurements in the direct sun mode as well as observations in the Umkehr mode. The lamp tests remain semi-automated since an operator still has to set the lamps in place before launching the automated recording of tests results.

This paper is published in parallel with the analysis of the LKO Dobson instruments data using the manual and automated modes side-by-side by *Stübi et al.* (2020). Therefore, only few measurement results will be presented here and the reader is
10 invited to look at this companion publication for more information. Similar studies with LKO Brewer instruments have also been published recently (*Stübi et al.*, 2017a, b).

The paper is organized as follows: in section 2, the Dobson instruments measurements principles are presented while in the different sub-sections of section 3 the details of the transition from manual to automated measurements are presented. Discussion and conclusion follow in section 4 and 5.


## 2 Dobson measurement principle and instrument design

The principle of the Dobson instrument, its operation and data handling are described in many publications (*Komyhr*, 1980; *Evans*, 2008; *Basher*, 1982). An illustration of the essential parts of a Dobson instrument can be see in figure 1 of *Evans* (2017). The measurement principle is based on the comparison of the direct sun irradiance in two narrow bands of the UV

radiation spectrum selected by a pair of slits: the narrower slit selects the short wavelength window, $\lambda_s$, largely attenuated by the ozone in the atmosphere while the wider slit selects the long wavelength window, $\lambda_l$, mostly unaffected by ozone. The detection is based on the comparison of the signals from each optical path, as the slits are alternatively opened and closed by a rotating wheel. Since the absolute measurements of small signal was quite challenging at the time of the development of the instrument, G. M. B. Dobson introduced a calibrated attenuator in front of the long wavelength slit to attenuate the intensity of

$\lambda_l$ to the level of the $\lambda_s$ intensity. The attenuator thus imprints the effect of ozone at $\lambda_s$ along the light path in the atmosphere at the non-absorbing wavelength $\lambda_l$. The measurements consist of adjusting the attenuator until the differential signal from the two slits is zero. For the Dobson instrument, the wavelength pairs are referred to as A, C and D, while their combinations are referred to as the double pair AC, AD and CD. The Dobson is a double monochromator instrument with a dispersing prism in front of the slits and a second prism after the slits to redirect the optical paths on the photo-multiplier detector. The optical

attenuator consists of a moving neutral density filter (the optical "wedge") attached to a graduated rotating disk (R-dial). As the slits are fixed to the frame of the instrument, the wavelength pair selection is achieved by rotating a pair of a quartz plates (Q1-lever, Q2-lever) through which the light beam passes.

The ozone column density is related to the measured irradiance intensity at the surface according to the Beer–Lambert law (*Moeini et al.*, 2019):

$$I(\lambda) = I_0(\lambda) \cdot e^{-\tau(\lambda)} \quad with \quad \tau(\lambda) = \alpha(\lambda) \cdot X \cdot \mu + \beta(\lambda) \cdot \frac{p}{p_0} \cdot m_R + \delta(\lambda) \cdot m_M \quad (1)$$

where $I_0(\lambda)$ is the extraterrestrial irradiance, and the optical thickness $\tau(\lambda)$ of the incident path is the sum of the ozone absorption, the Rayleigh and the Mie scattering terms. The coefficients $\alpha(\lambda)$ and $\beta(\lambda)$ are calculated based on the nominal values of the optical characteristics of the primary reference instrument (*Komhyr et al.*, 1989). From the calibration response of the attenuator, the R dial reading is converted to the relative intensity $N(\lambda_s, \lambda_l)$ for a pair of wavelength as $N = \log(\frac{I_{s0}}{I_{l0}}) - \log(\frac{I_s}{I_l})$.

Subsequently, the double pairs are formed allowing the elimination of the small contribution of the Mie scattering since it is almost constant in the $\lambda_s - \lambda_l$ range. Table 1 gives the nominal optical characteristics of the primary reference Dobson as well as those of the LKO instruments (*Komhyr et al.*, 1989). Even though the nominal and the actual values of the slit center wavelengths agree well, significant differences exist for the full-width-at-half-maximum (FWHM) values. Similar differences have been reported in *Köhler et al.* (2018) for different Dobson instruments characterized during the ATMOZ project (*ATMOZ*,

2018). The authors have quantified the effects of these differences on the calculated ozone column and conclude that for the commonly used double pair AD, the ozone values could be biased up to ∼1% depending on the instrument. Future advanced data reprocessing will incorporate these recent slits measurements for accurate comparison between different types of instruments.

The Umkehr method allows to estimate the vertical ozone profile from the measurements of relative intensities $N(\lambda_s, \lambda_l)$ of





**Table 1.** Dobson instruments nominal values of $\lambda_s$ [nm] and $\lambda_l$ [nm] center lines and FWHM as well as the equivalent values for the LKO Dobson instruments.

| Pair | Nominal $\lambda_s/\lambda_l$ | $D_{101}$ $\lambda_s/\lambda_l$ | $D_{051}$ $\lambda_s/\lambda_l$ | $D_{062}$ $\lambda_s/\lambda_l$ | Nominal FWHM | $D_{101}$ FWHM |
|------|------|------|------|------|------|------|
| | $\lambda_s$ [nm]/ $\lambda_l$ [nm] | $\lambda_s$ [nm] / $\lambda_l$ [nm] | $\lambda_s$ [nm] / $\lambda_l$ [nm] | $\lambda_s$ [nm]/ $\lambda_l$ [nm] | $\lambda_s$ [nm] / $\lambda_l$ [nm] | $\lambda_s$ [nm] / $\lambda_l$ [nm] |
| A | 305.5 / 325.5 | 305.6 / 325.4 | 305.6 / 325.2 | | 0.9 / 2.9 | 1.2 / 3.5 |
| C | 311.5 / 332.4 | 311.7 / 332.6 | 311.5 / 332.5 | 311.5 / 332.9 | 0.9 / 2.9 | 1.2 / 3.7 |
| D | 317.6 / 339.8 | 317.7 / 340.0 | 317.6 / 340.0 | 317.6 / 340.5 | 0.9 / 2.9 | 1.2 / 4.0 |

zenith observations at sunrise and/or sunset. As the solar zenith angle (SZA) is increasing, the two intensities $\lambda_s$ and $\lambda_l$ decrease, $\lambda_s$ more rapidly than $\lambda_l$. When the effective scattering height for the shorter wavelength $\lambda_s$ is above the ozone layer, its intensity decreases less rapidly because of the absorption occurs mostly after the scattering event. The N($\lambda_s$, $\lambda_l$) measurements presents a maximum as illustrated in Figure 4 before 6 o'clock and this curve bending is called the Umkehr effect (*Mateer*, 1964). The ozone profiles from ground to 50 km with a vertical resolution of 5-10 km is retrieved from the N($\lambda_s$, $\lambda_l$) curve by an optimal estimation method (*Petropavlovskikh et al.*, 2005).





## 3 Automation of the LKO Dobson instruments

In 1988, with the introduction of a rotating cabin, a major revision of the LKO measurement setup took place that greatly simplified the manual operation of the Dobson instruments as illustrated on the left side in Figure 1 (*Hoegger et al.*, 1992; *Staehelin et al.*, 1998). The rotating cabin brought an improvement of the data quality and reproducibility :

– A three minutes cycle was sufficient to perform the C-D-A measurements with two instruments;

– Ten second averages of the R-dial position and temperature could be recorded digitally;

– The instruments were nor longer exposed to adverse conditions like large temperature changes;

– The rapid start up permits to capture short sunny periods in changing weather conditions.

During the same period, a first Brewer instrument ($B_{040}$, MKII) was acquired to expand the fleet of instruments and possibly
make a transition to a partly or fully automated ozone monitoring station. Two other Brewer instruments were subsequently purchased in 1991 ($B_{072}$, MKII) and 1998 ($B_{156}$, MKIII) and operated in parallel with the Dobson instruments $D_{101}$ and $D_{062}$ (*Stübi et al.*, 2017a).

The first attempts at Dobson automation were made in the 1970s to reduce the effort for the Umkehr measurements that require to start in the morning before sunrise until the solar zenith angle reaches SZA = 60°, and to restart the measurements in
the evening from SZA = 60° till after sunset. *Räber* (1973) describes this first LKO realisation of a Dobson automation. While it was successful for the Umkehr measurements, the results were not conclusive for direct sun ozone column observations. In 2010, the decision was made to develop a fully automated version of the Dobson instrument. The conditions were to leave the internal optical parts of the instruments untouched to avoid any change in the measurement principle and to still allow manual operation if necessary. The result of this effort is shown in Figure 1.

The right side of Figure 1 shows the interior of the container with two automated Dobson side by side on lifting table. The Dobson instrument, the data acquisition system and the computer are all placed on a large aluminium plate that is mounted on a turntable which follows the sun azimuth. The solar radiation enters the instrument via the sun-director prism that protrudes the roof of the container and is protected from adverse weather conditions by a quartz dome.

The automation was realised on the Dobson instruments $D_{051}$ and $D_{062}$ between the inter-comparisons 2010 and 2012.
Dobson instrument $D_{101}$ was kept manually operated until beginning of 2014. The extended 2012–2014 development period allowed us to compare the manual and automated measurements (*Stübi et al.*, 2020).





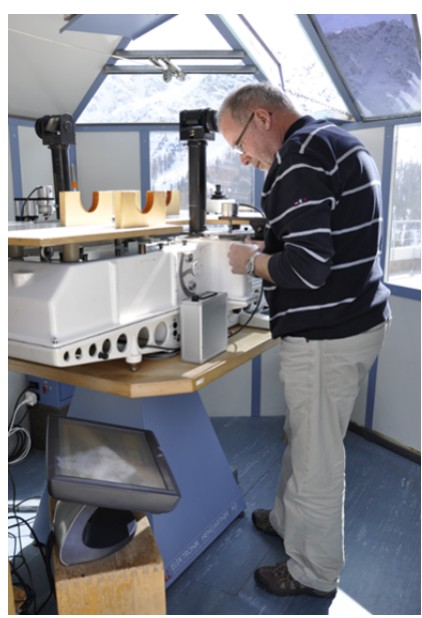
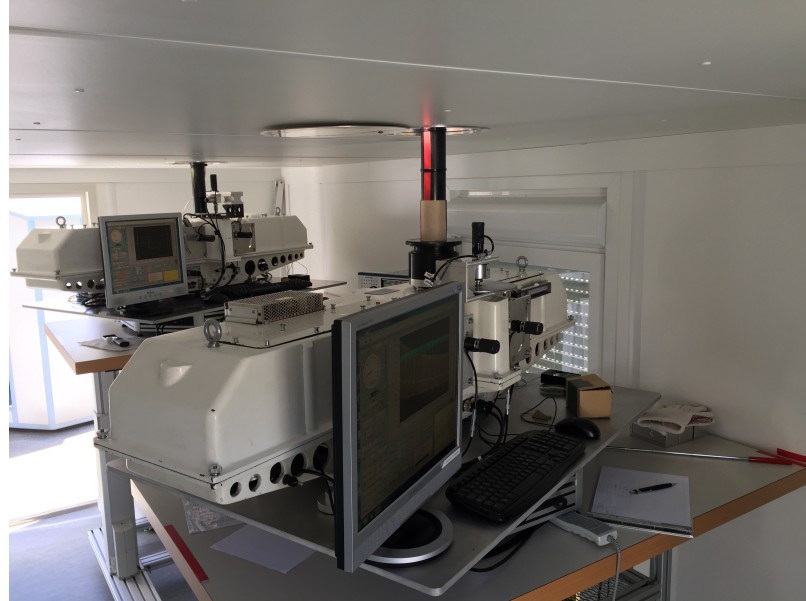

**Figure 1.** Illustration of the transition from manual to automated Dobson measurements. Left picture: operator in the rotating cabin with two Dobson instruments on a turntable and with an open skylight in the roof. Right: two automated Dobson instruments each on its rotating table.

## 3.1 Instrument Control

The Dobson standard operating procedure of the WMO recommended to measure the ozone column at selected SZA values to cover the air mass range $1 < \mu < 4$. This yields around 20 measurements in summer clear sky days but only a few measurements in winter. The automation of the LKO Dobson instruments allows to record the regular sequence of the three wavelengths pairs
5 C, D and A during the whole day starting (ending) when the sun passes above (below) the horizon at the station. With the operational parameters, a C-D-A measurement sequence takes typically 150 seconds. At the Arosa alpine site, the sun is above the horizon for 4.5 hours in winter representing ∼100 measurements for a sunny day, while in summer up to 250 measurements are recorded.

10 The Dobson instrument measurements require the control of 5 rotational axes:

- As for all direct sun measurements, the light entrance in the instruments requires to follow the azimuth and the elevation of the sun over the day. This require two rotation axes.

- The synchronous rotation of the two quartz plates to select the appropriate wavelengths pair C, D or A involves two more synchronised rotational axes.

15 - The R-dial rotating disk that drives the calibrated optical attenuator wedges is the fifth axis.





The calculated azimuth of the sun is followed by the rotating table (www.dr-clauss.de), which is driven by a stepping motor and controlled by a differential encoder with a resolution of 0.05°. Similarly, the sun elevation is controlled by an absolute encoder/motor small assembly (en.robotis.com) directly mounted on the sun director support. This device controls the prism orientation with a belt. The three other axes are driven by brushless DC motors and encoders co-axially mounted on the Q1 and

Q2 levers and on the R-dial. The data acquisition and control system are based on commercially available National Instrument (NI) components and the NI LabView programming language (www.ni.com). The interface between the NI motion control device (part NI-7340) and the motors (www.maxongroup.ch) and encoders (www.baumer.com) is realized with a commercially available controller box (www.sci-consulting.ch).

The controlling software was developed by Sci-Consulting Ltd. (www.sci-consulting.ch) in close cooperation with Me-

teoSwiss. The software architecture consists of micro-sequences that are called by a sequencer defining the chain of operations to fulfill the dedicated tasks. Figure 2 displays a chart of the major sequences of the data acquisition program: the left side refers to the automated operation for the direct sun, and the Umkehr measurements. The right side refers to the semi-automated standard lamp and Hg lamp tests. Following the initialisation of the daily files and referencing the encoders, the system waits for the calculated time of the sunrise above the station horizon. After setting the Q-levers for the first wavelength pair and

reading the initial R-dial position $R_{init}$ and the photomultiplier (PM) high voltage (HV) from the configuration file, the system adjusts these initial values to the actual conditions with the "3-points test". This latter consists of measuring the PM signal dlaP (**d**igital **l**ock-in **a**mplifier in **P**hase with the selector wheel) (5 seconds average) for three successive R-dial positions defined as:

$$R_1 = R_{init}, \qquad R_2 = R_1 \cdot (1 + sign(dlaP_1) \cdot 0.05), \qquad R_3 = R_2 \cdot (1 + sign(dlaP_2) \cdot 0.05) \qquad (2)$$

From the three pairs of points ($R_i$, $dlaP_i$), the R value corresponding to dlaP≈0 is inter- or extrapolated depending on sign(dlaP). The "3-points test" allows to measure the sensitivity of the PM response $\delta(dlaP)/\delta(R)$ as a Dobson operator would do with small back and forth R-dial rotations to adjust the HV. The data acquisition program similarly adapts the HV (within preset high/low limits) or repeats the "3-points test" until the PM response is above a predefined threshold. The upper panel of Figure 3 shows a few "3-points test" results: the R-dial positions follow the yellow line (left scale) and the PM dlaP re-

sponse the corresponding blue line (right scale). The test results are recorded as housekeeping information for quality control purposes. Once the R value for the dlaP≈0 condition is found, a proportional–integral–derivative (PID) controller acts on the R-dial position to maintain the PM signal close to zero for an averaging period of 20 seconds. Then the loop restart with the next wavelength until completion of a C-D-A cycle. All the parameters controlling the timing or threshold conditions of the measurement procedure are stored in a configuration file and can be adapted to local conditions. This setup ensures a large

flexibility of the data acquisition and instrument control program.

Umkehr measurements follow the same sequences as direct sun measurements except that it is not necessary to follow the sun elevation since the Dobson instrument points to the zenith. Umkehr data acquisition parameters are adapted to the lower light intensity, the different start/end of measurements conditions, etc.

The standard lamp test procedure illustrated in the right part of figure 2 is also similar to direct sun observations except that the sun pointing routines are skipped. For the Hg lamp test, the Q1 lever is moved at a predefined interval to scan the Hg spectral line at 312.96 nm. The PM dIaP values are averaged for 5 seconds at successive Q1 positions. Examples of the results of these two tests are illustrated in Figure 6 and presented in sub-section 3.4. The operator has to set the lamps in place and let

5 the program record around 10 C-D-A cycles of standard lamp measurements, respectively around ten scans of the Hg lines for stable and reproducible results.

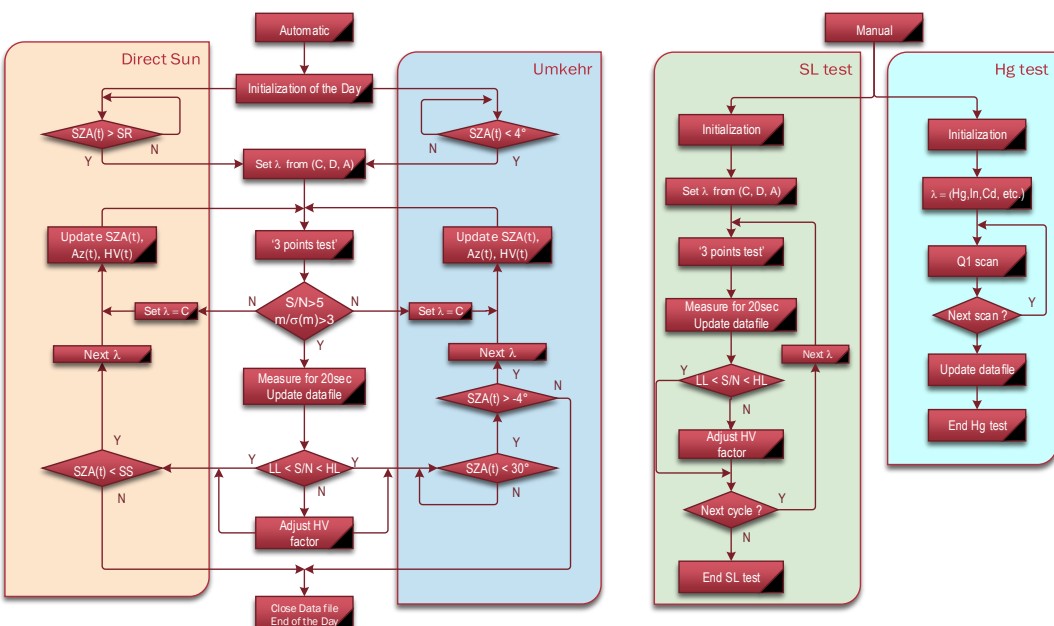

**Figure 2.** Schema of the data acquisition program's main steps for direct sun, Umkehr, standard lamp and Hg lamp measurements.





## 3.2 Photomultiplier signal treatment

The signal amplifier board for the photo-multiplier signal modulated by the selector wheel rotation was updated to use recent electronic components. It has been designed around a current amplifier (OPA129U) directly connected to the photo-multiplier output. There is a low pass filter with a cutoff frequency of ∼300 Hz to mitigate the Nyquist folding. The data acquisition

system records the selector wheel position indicated by a photo-diode and the amplified PM output AC signal is analysed with a digital look-in amplifier (dla), essentially by extracting the Fourier series coefficient at the frequency of the selector wheel.

A dynamical control of the system is achieved via a Fast Fourier Transform (FFT) analysis (frequency, signal/noise) to adapt the data acquisition parameters continuously. The bottom part of Figure 3 shows an example of the FFT of the dlaP (pink line), where the vertical line indicates the measured selector wheel frequency. The horizontal line indicates the median noise

level calculated in the frequency range 10–200 Hz. This is mainly generated by the PM HV and it is 2-3 orders of magnitude higher than the data acquisition noise level (yellow). The FFT is calculated on blocks of data (sampling rate 60 kHz) whose size corresponds to a multiple of the selector wheel period to improve the noise rejection. The system feedback loop acts on the R-dial motor to maintain the PM signal as low as possible.

The PM high voltage values for a few SZA values are defined in the configuration file and linearly interpolated in between. It is

adapted to the actual measuring conditions using the noise level measurements: if the noise falls below a lower limit, the HV is increased by 5%, and if the noise exceeds an upper limit, the HV is decreased by 5%. The adaptation factor is limited to ± 10% of normal value. With this controlled noise level conditions, the "3-points test" gives a measure of the PM signal sensitivity with the ratio $\delta(\text{dlaP})/\delta(\text{R})$. Accurate measurements can only be realised if the signal to noise ratio is ≥1. This parameter can thus be used to control the measurements in real time and/or for the off-line quality control of the data.

Fast changing sun irradiance associated to clouds is difficult to deal with. Bright sun can suddenly be completely blocked by a thick cloud, decreasing the PM signal by several orders of magnitude. A too fast increase of the PM high voltage would result in a saturated PM when the sun reappears. To avoid an oscillation of the R-dial feed-back, the acquisition parameters had to be determined empirically. For example, the 5% step change of the HV values and limited to ± 10% has been adequate for scattered clouds conditions.



**Figure 3.** Screen capture of the real-time data acquisition system since this information is not recorded. Upper panel : time series of the R-dial (yellow line) and the digital look-in amplifier signal dlaP (blue line) with the distinct "3-points tests". The green line is from a luxmeter instrument pointing to the zenith. Lower panel: FFT of the dlaP signal with the chopper frequency ($\sim 29$ Hz) marked by the vertical blue line, respectively the average noise ($\sim 10^{-4}$) between 10 Hz and 200 Hz by the horizontal blue line.



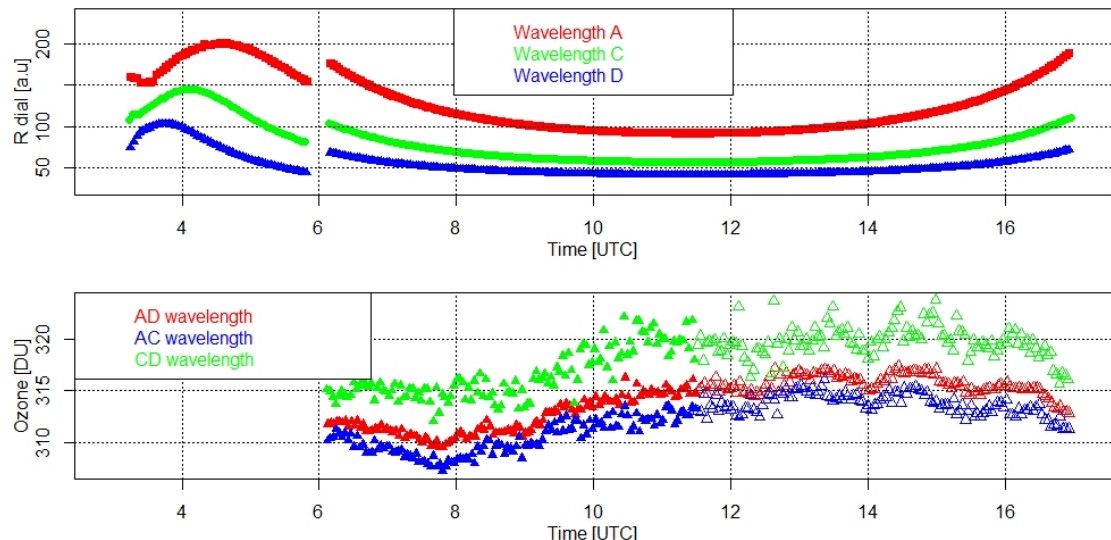

**Figure 4.** Time series of the measurements for 7 July 2020 with Dobson instrument $D_{051}$. Upper panel : time series of the R-dial positions. Lower panel: corresponding ozone column values for the double pairs AD, AC and CD. Filled symbols correspond to the morning, and open symbols to the afternoon time period with respect to local noon.

### 3.3 Measurements results

Figure 4 illustrates the results of the automated Dobson instrument $D_{051}$ measurements for the sunny day of 7 July 2020. The R-dial positions for three wavelengths pairs C, D and A are shown in the upper panel, starting with the Umkehr measurements until 6 am and direct sun observations for the rest of the day. The corresponding ozone columns for the double pairs AD, AC and CD are shown in the lower panel. The reference AD ozone column showed low point-to-point differences of $\leq 1$ DU and a smooth variation during the day. The AC double pair showed a systematic low bias of $\sim 2$ DU compared to the AD double pair and twice larger point-to-point fluctuations. The CD double pair ozone column was 2% higher than the AD and a scatter of $\sim 2\%$. The standard deviation of the R-dial for 20 second averages were $\leq 0.2$ degree in such sunny conditions and were therefore smaller than the symbols used in the figure.

The Umkehr data are also very smooth and show clearly the wavelength dependent "Umkehr" points where the various R-dial curves exhibit their maxima. The switch-over from Umkehr to direct sun observation for Dobson instrument $D_{051}$ was done manually since the option to remove the sun director device has not been automated up to now.

Figure 5 illustrates the housekeeping information measured in parallel to the measurements of Figure 4. The noise level deduced from the FFT analysis of the measurements in the upper panel shows high values due to the high voltage of the PM necessary to measure the very low intensity of the zenith scattered light at sunrise. Correspondingly, on the lower panel the signal-to-noise ratios are close to 1 initially and gradually increase afterwards. After 6 am, the direct sun measurements present better signal-to-noise ratios. A transition happened at 11 am, when the noise level of the A wavelength pair dropped below the





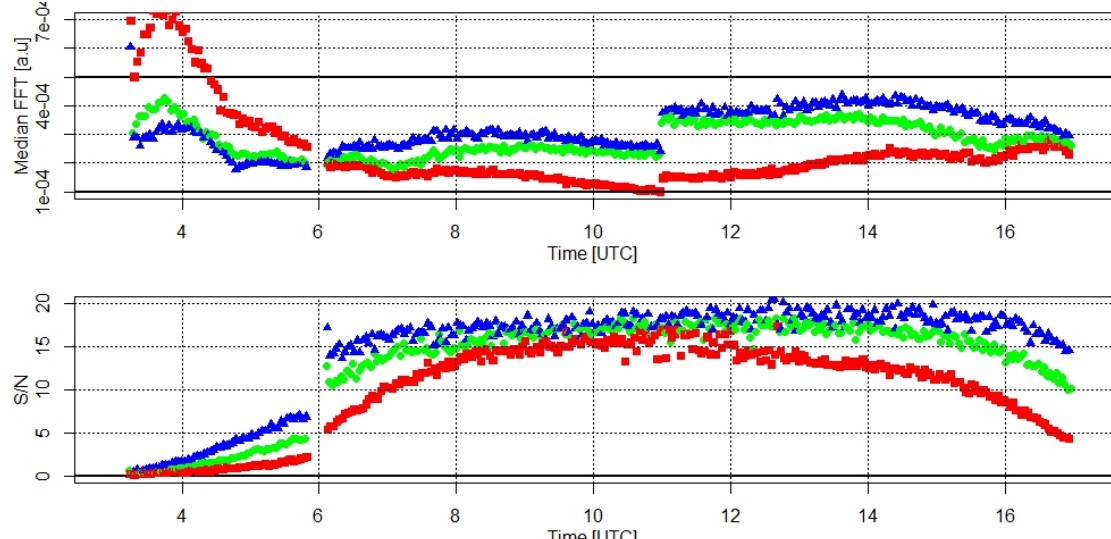

**Figure 5.** Time series of the housekeeping parameters corresponding to the measurements of Figure 4 with the same color coding. Upper panel: time series of the median value of the FFT transform integrated between 10 and 200 Hz. The horizontal black lines correspond to the high and low limit controlling the HV corrections. Lower panel: time series of the signal-to-noise ratio S/N = $|(\delta(\text{dlaP})/\delta(\text{R}))|$ / median($\int FFT(\lambda)d\lambda$).

lower limit (lower black line), and the HV of the PM was increased by 5% of the prescribed values. It stayed the same for the rest of the day.



### 3.4 Automation of instrument tests

Once the measurements procedure has been developed, the data acquisition (DAQ) system can be programmed for realising specific operations. The more common ones are the standard lamp and Hg lamp tests performed weekly to assure the stability of operation of Dobson instruments. As illustrated in Figure 2, the standard lamp test procedure is similar to the direct sun

measurements with no control of the sun position and different settings for the HV and R$-init$ values. The upper panel of Figure 6 shows the results of standard lamp tests using successively 3 different standard lamps. The C and D wavelength pairs R-dial values were very stable to within ≤0.05 degree while the A pair results varied within ≤0.1 degree. The duration of the tests is left to the operator's judgement but usually a dozen points are sufficient to assure stable conditions. The mean R-dial values for each wavelength and for each standard lamp are supposed to stay within 0.3 degree on the mid- to long-term for

stable instruments. Larger changes are normally a sign of either an aging lamp or a change in the instrument response and are corrected by an update of the attenuator calibration curve.

The Hg lamp test is performed to check the optical alignment of the Dobson instrument. The test consists of scanning the 312.96 nm Hg spectral line through the slits S2 by moving the Q1-lever wavelength selector (test S2Q1). The lower panel of Figure 6 shows the results of a Hg S2Q1 test with a Gaussian fit of the measurements. The calculated central value was at 313.0

nm in good agreement with the Hg spectral line while the full-width-at-half-maximum FWHM = 1.4 nm is slightly larger than the 1.2 nm measured values of Dobson instruments $D_{101}$ given in table 1. In general, the tests consist of a succession of 10-20 scans to assure a stable Hg lamp temperature. A deviation of the Hg test results from the nominal values require an adaptation of the Q1 setting table and/or its temperature correction factor. Similarly, the test can be done by moving Q2 instead of Q1 lever (test S2Q2) or having the light path through slit S3 instead of S2 (tests S3Q1 and S3Q2). Significant difference between

the central position determined in each test is the sign of an optical misalignment of the instrument.

Besides the lamps tests presented above, which require the presence of the operator once a week, the DAQ system affords the possibility to check the behavior of the instrument remotely. One of the key functions of the instrument control program is tracking the sun position, so two tests were implemented to scan the azimuth and elevation of the sun. Figure 7 illustrates the results of these tests: the dlaP signal and the noise level were measured (5 second averages) at different azimuth (upper

panel) or elevation (lower panel) angles. On both graphs, the actual values are marked by the vertical blue lines that lie on the plateau part of both the dlaP and noise curves. These 2 minute tests were realised close to local noon so that the R-dial position could stay constant. At other times of the day, it is necessary to correct for the change of the R-dial value. The low values of the FFT noise level at both ends of the tests indicate that the entrance slit was not illuminated by the sun, while the flat part of the curves in the center of the figures indicate full illumination. The parameters of these tests (averaging period, number of

positions, scan interval, etc.) which can be adapted to local measurements conditions are stored in the configuration file.

Another test known as the "S-curve test" was described by G. W. Dobson in the Dobson instrument reference manual (revised version) (*Evans*, 2008). It allows to check the optical alignment of the instrument by scanning the sun irradiance around the nominal values of the three wavelength pairs. From the shape of the ozone absorption cross-section function, a symmetric response of the R-dial position is expected. A variation of this test was implemented in the DAQ system: the wavelength pairs



**Figure 6.** Results of semi-automatic standard lamp, respectively Hg lamp tests of Dobson $D_{062}$ instrument; Upper panel: R-dial record of three different standard lamps for the wavelength pairs A (green), C (red) and D (blue). Lower panel: PM dlaP response to the Q1 scan of the Hg lines at 312.96 nm: measured values (blue), fitted curve (red) and residuals (green).





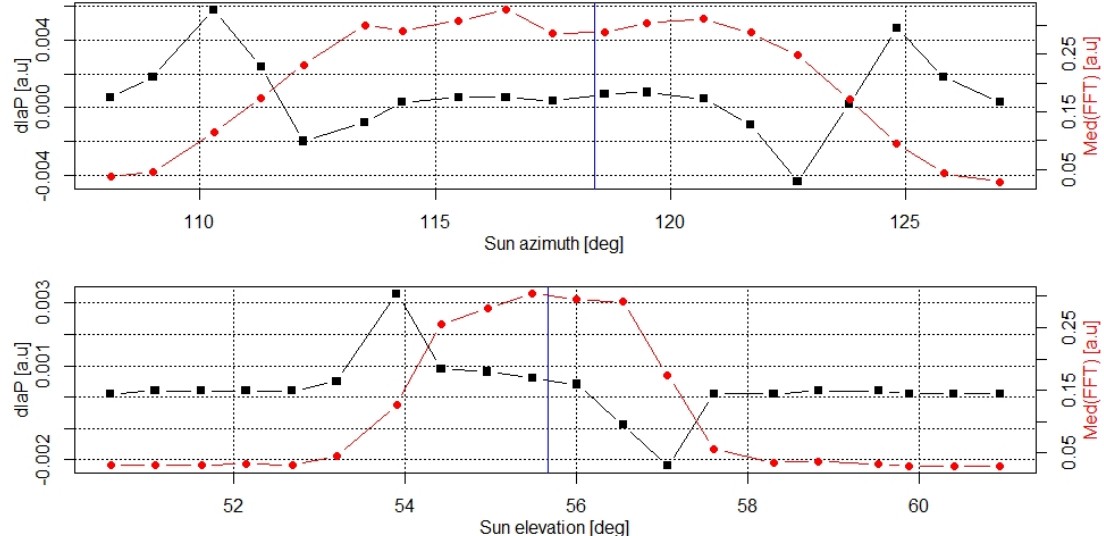

**Figure 7.** Results of the tests to control the sun azimuth and elevation. Upper panel: dlaP signal (black) and FFT noise level (red) for a 20 degree sun azimuth scan. The vertical blue line indicates the nominal value for a direct sun measurement. Lower panel: similar results for a 10 degree sun elevation scan.

are scanned simultaneously around the C, D and A nominal values. Around local noon, the R-dial positions corresponding to dlaP≈0 do not change. Therefore from the locally linear relationship between the dlaP signal and R-dial position, scanning simultaneously the Q1 and Q2 levers produces a "S-curve" response. G. W. Dobson reports the difficulty to obtain a nice "S-curve" for the C-wavelength pair and also the limited scanning range accessible for the D-wavelength pair. Examples of these

5    "S-curves" are shown in Figure 8 for the three wavelengths pairs.

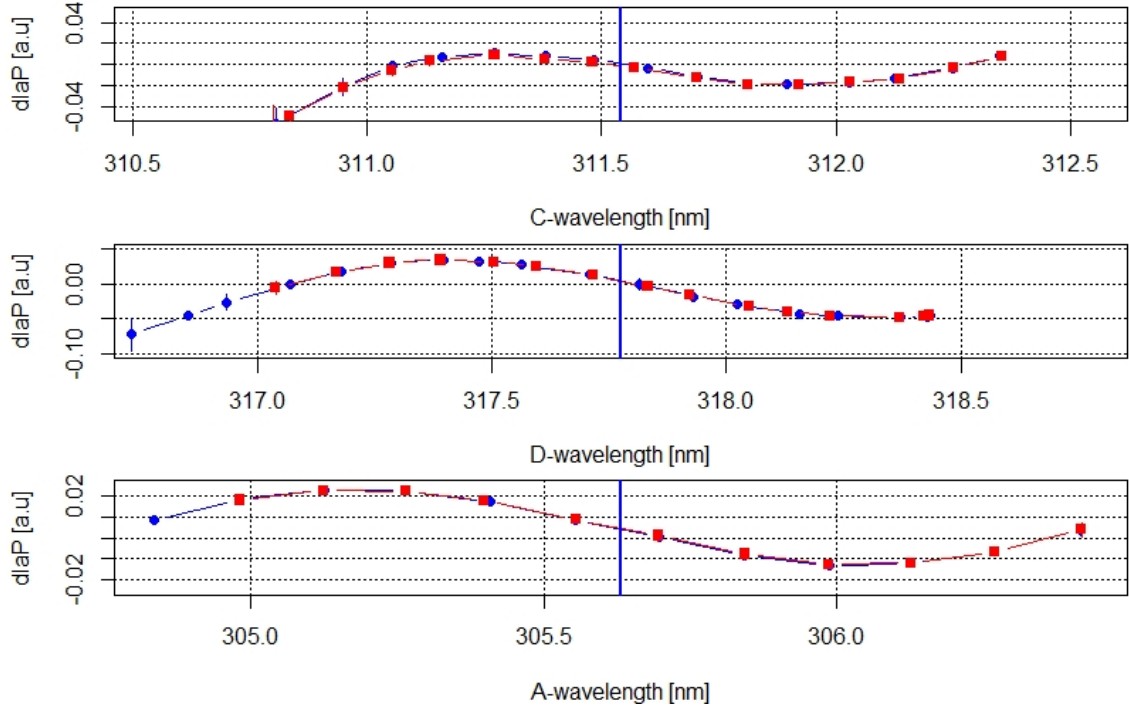

**Figure 8.** S-curve test results for the three wavelength pairs C (two scans), D and A. Nominal wavelengths are marked with the vertical blue lines at 311.5 nm, 317.7 nm and 305.6 nm.

## 4 Discussion

The automated method developed at MeteoSwiss has brought great benefits and flexibility to the use of the Dobson instrument. Besides the improvement of the data quality and measurement frequency discussed extensively in the separate paper by *Stübi et al.* (2017a), the lifetime of these already old instruments will be extended for years if not decades. This is an important

5   perspective for long term monitoring of climate change related parameters like the ozone column.

The continuous data recording allows to follow relative daily variations as small as a fraction of a percent, which opens up possibilities to study variations on a time scale and with a measurement quality never achieved before with the Dobson instrument. The reproducible automated measurements could be used to develop advanced data treatment algorithms taking into account the individual optical characteristics of the instruments. This could help to identify the origin of the systematic

10  biases between the double wavelength pairs or to refine the treatment of the Rayleigh and/or Mie scattering terms.

The traditional standard lamp and Hg lamp tests could be fully automated by adapting a mechanism to slide the lamps in the sun director support. Such an option could be useful for remote sites like in the Arctic or Antarctic that are inaccessible for part of the year or to alleviate operational budget restrictions. Similarly, the option of automatically changing from zenith (Umkehr) to direct sun measurements could be automated using commercially available robotic arms to remove the sun director support.





Presently most inversion algorithms for the Umkehr data to estimate the ozone profiles between 20 and 60 km altitude use only the C wavelength measurements. An algorithm to include the combined information of the three wavelengths has been developed recently (*Stone et al.*, 2014). The improved reproducibility and quality of the automated Dobson instrument data associated to the measured uncertainty could be beneficial to improve the Umkehr inversion algorithm.

5    Other Dobson instrument partial or full automation projects have occurred since the 70s (*Räber*, 1973; *Malcorps and de Muer*, 1977; *Miyagawa*, 1996; *Koomhyr et al.*, 1985; *Kim et al.*, 1996). Unfortunately, comprehensive description of these developments in English are generally missing and difficult to find in the open literature. Therefore, a detailed comparison with the data acquisition scheme presented above is not feasible. The automated Dobson system developed by *Miyagawa* (1996) is the most widespread, in particular it has been installed at the NOAA's Dobson network stations (*Evans*, 2017). Since Japan is

10    discontinuing their operational Dobson network, probably no major development of this system will occur. The particularity of the Swiss system is that the instruments are kept in a container with stabilized temperature and the use of a quartz dome. This option prevents the exposure of the instrument to outside ambient conditions as happened by opening of a large dome or moving the instruments out of his shelter as most other systems do.





## 5   Conclusion

The automation of the control and data acquisition of Dobson sun spectrophotometers at MeteoSwiss was a challenging development over the course of several years. Three Dobson instruments have been running in automated mode in Arosa and Davos without major problem for 5 years since the completion of the development phase. The short C-D-A measurement cycle of ∼150 seconds allows to follow ozone column variation at the level of ± 1 DU in clear sky conditions. The housekeeping data produced by the system has proven useful for the data quality control and may facilitate further understanding and improvements of these instruments and the inversion algorithms of Umkehr data in the future. In the associated paper by *Stübi et al.* (2017a), the results of the detailed analysis of the Arosa ozone column measurements with three automated Dobson instruments is presented.

## 6   Data availability

Operational ozone column data from Dobson D101 until 2014 and D062 since 2014 are available at WOUDC and NDACC. For the extended set of housekeeping data, contact the corresponding author.

*Author contributions.* RS has made the analysis of the data and written the first version of the manuscript. HS was in charge of the quality control and the preparation of the data sets. EMB, JK, HS and AH have contributed to the discussions and revisions of the manuscript.

*Competing interests.* No competing interests.

*Acknowledgements.* We would like to thank the PMOD/WRC staff for their great support to run our instruments on their premises and for the excellent collaboration.



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
