# Peer review of "A fully Automated Dobson Sun Spectrophotometer for total column ozone and Umkehr measurements"

_Atmospheric Measurement Techniques, 2020_

## Referee Comment (RC1) · Anonymous Referee #2 · 31 Dec 2020

A fully Automated Dobson Sun Spectrophotometer for total column ozone and Umkehr measurements René Stübi, Herbert Schill, Jörg Klausen, Eliane Maillard Barras, and Alexander Haefele

Initial Comments: This submitted manuscript would fit in the category of a commentary, as it describes the modernization of an existing measurement program. I recommend publication after the issues below are addressed. Technical corrections are in the supplement file

Specific Comments/Questions:

I am not commenting on the details of the automation and electronics, as I am not

currently experienced in this field.

Introduction: A reference to the history of the discovery of ozone depletion by chlorofluorocarbons (CFCs) is expected. The discovery of the Antarctic ozone hole by Dr. Solomon (https://doi.org/10.1038/d41586-019-02837-5) is a good source for the history. Suggest a lead-in sentence similar to: The history of the detection of the ozone layer depletion is one of the most important scientific stories of the 20th century (Solomon, 2019). Line 20: The term "calibrate" is not correct. This paragraph should be re-written. The various ground-based and space-based networks have independent calibration methods. Data results from the various instruments and networks are inspected and intercompared to detect problems within networks. I believe that authors are also saying that the algorithms used to convert data to total ozone values are evolving with increased understanding of the instrument characteristics, and the assumptions used in the measurement and data reduction algorithms. Line 25: There should be some mention of the development of the instrument for the early 1900s' studies in Stratospheric Circulation.

Dobson measurement principle and instrument design: Table 1: Why is there no A-pair wavelength values for D062 or FWHM values for D062 or D051?

3.4 Automation of instrument tests There are other processes in the operation of a Dobson observing program. One of which is determining the attenuation curve of the optical attenuator. Has there been an attempt to automate this process?

Page 14, line 10: Larger changes are normally a sign of either an aging lamp or a change in the instrument response and are corrected by an update of the attenuator calibration curve. A better explanation is required. The data reduction algorithm incorporates the changes in the standard lamp test values from the lamp values determined at the time of the instrument's calibration by comparison to a reference instrument. The attenuator calibration curve is determined by a different procedure. The standard lamps are actually reference lamps, with measured values for a certain

Dobson instrument on a specific date. The change in the measured values with time indicates aging of the instrument. Use of multiple lamps on varying time schedules allows for detection of aging lamps.

Please also note the supplement to this comment:
https://amt.copernicus.org/preprints/amt-2020-391/amt-2020-391-RC1-supplement.pdf
* * *
[Figure]

**Supplement:**

**Review amt-2020-391**
**A fully Automated Dobson Sun Spectrophotometer for total column ozone and Umkehr measurements**
René Stübi, Herbert Schill, Jörg Klausen, Eliane Maillard Barras, and Alexander Haefele

**Initial Comments:** This submitted manuscript would fit in the category of a commentary, as it describes the modernization of an existing measurement program. I recommend publication after the issues below are addressed.
* * *
**Specific Comments/Questions:**

I am not commenting on the details of the automation and electronics, as I am not currently experienced in this field.

**Introduction:**
A reference to the history of the discovery of ozone depletion by chlorofluorocarbons (CFCs) is expected. The discovery of the Antarctic ozone hole by Dr. Solomon is a good source for the history. Suggest a lead-in sentence similar to: *The history of the detection of the ozone layer depletion is one of the most important scientific stories of the 20th century (Solomon, 2019).*
Line 20: The term "calibrate" is not correct. This paragraph should be re-written. The various ground-based and space-based networks have independent calibration methods. Data results from the various instruments and networks are inspected and intercompared to detect problems within networks. I believe that authors are also saying that the algorithms used to convert data to total ozone values are evolving with increased understanding of the instrument characteristics, and the assumptions used in the measurement and data reduction algorithms.
Line 25: There should be some mention of the development of the instrument for the early 1900s' studies in Stratospheric Circulation.

**Dobson measurement principle and instrument design:**
Table 1: Why is there no A-pair wavelength values for $D_{062}$ or FWHM values for $D_{062}$ or $D_{051}$?

**3.4 Automation of instrument tests**
There are other processes in the operation of a Dobson observing program. One of which is determining the attenuation curve of the optical attenuator. Has there been an attempt to automate this process?

Page 14, line 10: *Larger changes are normally a sign of either an aging lamp or a change in the instrument response and are corrected by an update of the attenuator calibration curve.* A better explanation is required. The data reduction algorithm incorporates the changes in the standard lamp test values from the lamp values determined at the time of the instrument's calibration by comparison to a reference instrument. The attenuator calibration curve is determined by a different procedure. The standard lamps are actually reference lamps, with measured values for a certain Dobson instrument on a specific date. The change in the measured values with time indicates aging of the instrument. Use of multiple lamps on varying time schedules allows for detection of aging lamps.
* * *
**Technical Corrections/Comments/Suggestions**

**Abstract:**
Line 3: Suggest: *However, the Dobson sun spectrophotometer requires manual operation which has led to the discontinuation of its use at many stations, thus disrupting long term records of observation.*
Line 8, Suggest: *Compared to manual operation, the automation results in a higher number of daily measurements with lower random error and additional housekeeping information to understand the measuring conditions.*

**Introduction:**
Line 9: Suggest: *Moreover, the uncertainties associated with climate change feedback on the ozone recovery process require dedicated ground-based measurement networks for sustained monitoring*.
Line 14: Suggest: *The principle of the instrument developed by G. M. B. Dobson in the early 1920s is based on measurements of the intensity of ozone-attenuated radiation in a number of narrow spectral bands. This was first done by analyzing spectra recorded on photographic plates, later directly on spectra within the instrument with photoelectric detectors and nowadays with photo-multipliers (PM) detectors.*
Page 2, Line 32: Suggest: *After the discovery of ozone layer depletion by CFCs, the measurement program continued as part of the global effort to verify that the Montreal Protocol was working. The more recent automation of the Dobson operation allows for continuation and improvement of this observation program, with reduced operational cost.*

**3 Automation of the LKO Dobson instruments**

Suggest starting with a sentence similar to:   *The instrument and observation facilities have had numerous improvements over the years.*

**3.1 Instrument Control**
Page 7, Line 7: the *local* horizon
Page 7, Line 11: Suggest: *The sun's image must fall on the entrance window of the instrument, thus the sun's azimuth and elevation must be tracked.*

**3.3 Measurements results**
Page 12, line 12: suggest: *has not yet been automated*.

**3.4 Automation of instrument tests**
Page 14, Line 2, Suggest: *Once the measurement procedure had been developed, the data acquisition (DAQ) system could then be programmed to perform other specific tasks*.

Page 14, Line 12: Suggest: *A Hg lamp is used to verify the wavelength settings and to check the optical alignment of the Dobson instrument.*

**Discussion**

Page 18. Line 6 **:** Komhyr not Koomhyr
Page 18, Line 10: Suggest: *The Swiss automation system is unique in that the instruments...*

**References:**
Page 20, Line 20 Komhyr, not Komyhr -- references in text (Page2, line 19; Page 4, Line 2)  have to be corrected

---

## Referee Comment (RC2) · Anonymous Referee #1 · 1 Mar 2021

General Comments

The authors describe an automated system for operating the Dobson spectrophotometer, which, despite its age, is still very widely used around the world for high-quality ground-based measurements of stratospheric ozone. The system described in the manuscript presents a large number of advantages and might be very attractive to Dobson operators in many countries. The authors should be commended for their effort to document the system in this manner, and in my view the manuscript is very welcome in AMT, subject to some minor revisions.

My only substantial comment on the manuscript is that in several places, the authors

could do better to explain how they reach their conclusions. At times, the authors seem to implicitly assume that the reader already knows what they are talking about, and a reader outside the Dobson community or even one not familiar with the Arosa Dobson station might have difficulty following the logic. One example is the list of benefits of the so-called "rotating cabin" (page 6), but a more serious one is that in several places it is stated that the automation leads to a higher "quality" of data, and that this will facilitate improved understanding of various Dobson issues, without much explanation to the reader of why this is.

One other point to raise is that, from my understanding of the system, it is not possible to make zenith observations during the daytime while operating automatically – is that correct? This seems to be a significant limitation of the system, compared to standard manual operation. This point should be clarified or discussed.

Specific comments

Line 2 "state of the ozone layer" would be better wording

Lines 2-3 Reword this whole sentence please

Lines 3-6 I would say the main point is that the calibration drifts during the time the satellite is in orbit

Line 6 – I don't understand what the authors mean by "developed over 50 years" – the instruments or the networks? People typically date the "Dobson network" to 1957, but sometimes much earlier.

Line 9 Why do you cite Pawson et al. (the 2014 WMO Ozone Assessment) and not the corresponding chapter from 2018? The 2018 assessment is cited in general but no specific chapters are referred to.

Line 10 "to" should be "with"

[Figure]

Lines 12-14 This statement is confusing because you start with wavelengths below 300 nm which you say are "almost completely absorbed" but then finish with saying the ozone column controls UV intensity at ground level – the reader should understand what wavelength range you're talking about and distinguish between UV-B and UV-C.

Line 17 I think this statement understates the age of a lot of the Dobsons in the network – many of the very old ones are still being operated, not just instruments made before the year 2000. (http://www.o3soft.eu/dobsonweb/instruments.html)

Lines 28-33 This is all interesting but I don't see that it's relevant. (You should still provide the references for the reader who wants to know more about the history of Arosa though).

Line 1 – "strengthening" should be "strengthen"

Lines 2-3 Was there any reason to think either that extra measurements would be beneficial, or that operator influence was having a non-negligible effect on the measurements?

Lines 5-8 It sounds though it is not possible to automatically perform DS/ZB or DS/ZC pairs of observations as is the practice at many stations, or to choose to take zenith observations during the day if DS is not possible for a time?

Line 3 – I think you really need a diagram of your own, showing the components of the Dobson that will be referred to in the rest of the manuscript.

Line 4 "direct sun" – not necessarily, in the case of Umkehrs or zenith observations.

Line 9 Just "Dobson" will do from now on, for the first mention you could give his initials.

Line 19 Moeini et al. 2019 seems a strange reference for the basic Dobson equation.

[Figure]

Line 22 The reader might not understand why these coefficents are based on properties of the primary reference instrument.

Line 26 I would prefer you give at least one more equation showing how the formula looks for four wavelengths rather than just one, so the reader can follow why the aerosol term is so small.

Line 27 If you are going to give these values, you should explain (very briefly) how you measured them particularly since the measurement of slit widths has been receiving a lot of attention in the Dobson community in recent years.

Line 34 "allows to estimate" – please re-word.

Line 1 If you say "at" sunrise or sunset the reader may have the impression it is a single measurement at that one moment, rather than over a period of time.

Line 4 Figure 4 is not the clearest diagram to explain the Umkehr effect but it's good enough I suppose.

Line 5 "is" should be "are"

Lines 4-8 You definitely need to explain this better. It would be very difficult for the reader to see why a "rotating cabin" would lead to these four improvements, especially the first two which might seem completely unrelated. What was the situation before the introduction of the cabin?

Lines 9-12 I don't see the relevance of this information?

Lines 15-16 I would like to see just a little bit more information about the earlier partial automation. What does "not conclusive" mean here?

Line 20 "table" should "tables"

Line 23 insert "from", ie "protrudes from the roof"

Lines 23 The quartz dome is not widely used around the Dobson network and needs a little bit of explanation, particularly about why you believe it does not interfere with the measurement.

Line 3 Why only 20 measurements?

Lines 4-9 You need to explain the reasoning here – why does the automation extend the valid mu range of the Dobson?

Lines 10-15 I really like the fact that you set out clearly a list of the five rotational axes.

Lines 1-8 This is probably a question more for the editorial team – I question the usefulness of including links to commercial websites. These will probably change within a few years. Specific information that appears on the website now is not guaranteed to still be there tomorrow.

Lines 1-5 How do you know that the tracking of the sun is sufficiently accurate? (I see some discussion of this point later on).

(Figure 4) The CD is more variable from point-to-point and interestingly, also has larger rises and falls than the AD and AC during the short-term variations. Would you like to comment on that?

Lines 11-12 It sounds like a person has to be at the site quite early in the morning to change from Umkehr to Direct Sun mode? This is disappointing from a pragmatic viewpoint because it reduces some of the benefit of automation.

Lines 10-12 Are you also performing sunset Umkehrs?

Lines 3-11 Are the results of the lamp tests over a long period of time able to be presented by the software in a user-friendly way? This would be extremely helpful, eg if you could track the lamp tests over a year or in between two calibrations.

Lines 25-30 This addresses the question I asked a couple of pages ago about the accuracy of the sun tracking. It sound like the check is not automatically scheduled but set into action by the operator? Are there any criteria for how close it needs to be? (The peak of the curve on the azimuth plot in figure 7 is quite wide). If the instrument levelling is not quite right of course the tracking might be correct at certain times but not others.

Page 16 Line 3 G.M.B. Dobson not G.W. I don't hesitate to say I believe Dobson would consider your S-curves to be very "nice".

Line 4 It's not clear how automating the "old" instruments extends their lifetimes? Do you mean just that they will be cheaper to operate, or something more?

Line 7 It's not clear how automating the instruments improves the measurement "quality". Especially in the "discussion" section you should expand on this point.

Lines 9-10 Similarly, you need to explain how automating the instruments will help address these longstanding issues.

Line 6 Correct the spelling of "Komhyr"

Lines 13 "his" should be "its"

Lines 4-5 This sentence is not very clear. Isn't the point the high-temporal-resolution,

not the +/- 1 DU? Is it useful to be able to follow these small-scale variations?

Lines 5-7 I think you could make it clearer to the reader why the housekeeping data would facilitate further understanding as you claim.

One question many readers would be interested in is what are the advantages and disadvantages of the automated Dobson system compared to a Brewer? (You might consider this to be out of scope for the current work, however).

Line 2 Albrecht is a strange choice – it's more about the policy issues of the Montreal Protocol

Line 5 I think you could do better here. The link only goes to the EURAMET page.

Line 1 Did you mean to refer to a particular chapter or chapters of the 2018 Ozone Assessment?

---

## Author Comment (AC1) · 13 Apr 2021

Reply to RC1 comments, AMT-2020-391 manuscript, "A fully Automated Dobson Sun Spectrophotometer for total column ozone and Umkehr measurements" by René Stübi et al."

The authors thank referee RC1 for the critical reading and the valuable comments and suggestions that allow us to improve our manuscript.

My only substantial comment on the manuscript is that in several places, the authors could do better to explain how they reach their conclusions. At times, the authors seem

to implicitly assume that the reader already knows what they are talking about, and a reader outside the Dobson community or even one not familiar with the Arosa Dobson station might have difficulty following the logic. One example is the list of benefits of the so-called "rotating cabin" (page 6), but a more serious one is that in several places it is stated that the automation leads to a higher "quality" of data, and that this will facilitate improved understanding of various Dobson issues, without much explanation to the reader of why this is.

=> The authors accept this remark that certainly is a bias found in many publications. It is also not clear if the referee has noticed that a companion paper containing more details on the Arosa station is under review in parallel to this technical paper. We have tried to improve the approach at various places in the manuscript besides those pointed out by the referee.

One other point to raise is that, from my understanding of the system, it is not possible to make zenith observations during the daytime while operating automatically – is that correct ? This seems to be a significant limitation of the system, compared to standard manual operation. This point should be clarified or discussed.

=> The statement is correct. Presently we have the chance in Arosa to have three Dobson instruments and so we dedicate one of them to Umkehr. Therefore, it was not in our priority to mount a robotic arm to remove the sun director for switching from direct sun to Umkehr observation and vice versa. In the discussion, we mentioned automated lamp tests and the adaptation of a robotic arm as useful extensions of the present system.

Specific comments

Line 2 "state of the ozone layer" would be better wording

=> corrected

Lines 2-3 Reword this whole sentence please

=> done

Lines 3-6 I would say the main point is that the calibration drifts during the time the satellite is in orbit

=> This is an additional point that we have introduced

Line 6 – I don't understand what the authors mean by "developed over 50 years" – the instruments or the networks? People typically date the "Dobson network" to 1957, but sometimes much earlier.

=> We refer to the network development. Since it was unclear, we rephrased this sentence.

Line 9 Why do you cite Pawson et al. (the 2014 WMO Ozone Assessment) and not the corresponding chapter from 2018? The 2018 assessment is cited in general but no specific chapters are referred to.

=> as suggested the reference to Pawson et al. is removed and chapters 3 and 4 of the 2018 assessment are mentioned

Line 10 "to" should be "with"

=> corrected

Lines 12-14 This statement is confusing because you start with wavelengths below 300 nm which you say are "almost completely absorbed" but then finish with saying the ozone column controls UV intensity at ground level – the reader should understand what wavelength range you're talking about and distinguish between UV-B and UV-C.

=> This sentence is now changed referring to the UV ranges UVA, UVB and UVC

Line 17 I think this statement understates the age of a lot of the Dobsons in the network – many of the very old ones are still being operated, not just instruments made before
the year 2000. (http://www.o3soft.eu/dobsonweb/instruments.html)

=> the reference to the second part of the 20th century is suppressed

Lines 28-33 This is all interesting but I don't see that it's relevant. (You should still provide the references for the reader who wants to know more about the history of Arosa though).

=> part of the sentences have been removed but the historical reference to LKO is still present

Line 1 – "strengthening" should be "strengthen"

=> corrected

Lines 2-3 Was there any reason to think either that extra measurements would be beneficial, or that operator influence was having a non-negligible effect on the measurements?

=> We had both reasons in mind. We were running for many years the LKO on a 365/365 daily schedule and we noticed differences between fulltime and part-time operators. Continuous measurements help the detection of bad/suspicious data and allow extending the measurements period at low sun elevation (high slant path) when the signal is very weak.

Lines 5-8 It sounds though it is not possible to automatically perform DS/ZB or DS/ZC pairs of observations as is the practice at many stations, or to choose to take zenith observations during the day if DS is not possible for a time?

=> This is correct and it returns to the specificity of the LKO favourable situation with no smog and rare hazy situations. However, the continuous observations by two Dobson pointing to the sun and one Dobson pointing to the zenith on a regular CDA cycle of less than 3 minutes cover partly the mentioned alternative observation types. The

objectives of the MeteoSwiss project were not the development of a general-purpose system even though the flexibility of the data acquisition program would allow it.

Line 3 – I think you really need a diagram of your own, showing the components of the Dobson that will be referred to in the rest of the manuscript.

=> We have asked the right to reproduce the diagram of Evans (2017) in a supplement Line 4 "direct sun" – not necessarily, in the case of Umkehrs or zenith observations.

=> corrected

Line 9 Just "Dobson" will do from now on, for the first mention you could give his initials.

=> corrected

Line 19 Moeini et al. 2019 seems a strange reference for the basic Dobson equation.

=> these references are primarily pointing to the Beer-Lambert law application. A reference to BAMS's Shaw publication is added.

Line 22 The reader might not understand why these coefficients are based on properties of the primary reference instrument.

=> a sentence is added to justify this choice.

Line 26 I would prefer you give at least one more equation showing how the formula looks for four wavelengths rather than just one, so the reader can follow why the aerosol term is so small.

=> the double pair AD equation has been introduced

Line 27 If you are going to give these values, you should explain (very briefly) how you measured them particularly since the measurement of slit widths has been receiving a lot of attention in the Dobson community in recent years.

=> A sentence to explain the slit width measurements is added with references.

Line 34 "allows to estimate" – please re-word.

=> corrected

Line 1 If you say "at" sunrise or sunset the reader may have the impression it is a single measurement at that one moment, rather than over a period of time.

=> corrected

Line 4 Figure 4 is not the clearest diagram to explain the Umkehr effect but it's good enough I suppose.

=> The Umkehr is not a central part of the publication, a separate figure is not essential

Line 5 "is" should be "are"

=> corrected

Lines 4-8 You definitely need to explain this better. It would be very difficult for the reader to see why a "rotating cabin" would lead to these four improvements, especially the first two which might seem completely unrelated. What was the situation before the introduction of the cabin?

=> an explanation of the setup before the rotating cabin and a reformulation of the improvements is added

Lines 9-12 I don't see the relevance of this information?

=> the information is moved in the introduction since it is effectively not relevant here.

Lines 15-16 I would like to see just a little bit more information about the earlier partial automation. What does "not conclusive" mean here?

=> explanation added

Line 20 "table" should "tables"

=> corrected

Line 23 insert "from", ie "protrudes from the roof"

=> corrected

Lines 23 The quartz dome is not widely used around the Dobson network and needs a little bit of explanation, particularly about why you believe it does not interfere with the measurement.

=> explanation added

Line 3 Why only 20 measurements?

=> it is not mandatory to follow the WMO recommendations (resulting to ∼20 data / summer day) that covers adequately the mu-range. Full time observers can do more observations while part-time observers may do only one observation.

Lines 4-9 You need to explain the reasoning here – why does the automation extend the valid mu range of the Dobson?

=> it is not directly mentioned that the mu range is extended but close to the sunrise /sunset, the signal are very weak and measurements are difficult. The automated digital data acquisition is pretty efficient when the signal/noise is close to 1.

Lines 10-15 I really like the fact that you set out clearly a list of the five rotational axes.

=> thanks for the comment

Lines 1-8 This is probably a question more for the editorial team – I question the usefulness of including links to commercial websites. These will probably change within a few years. Specific information that appears on the website now is not guaranteed to still be there tomorrow.

=> waiting for the editorial team response before editing

Lines 1-5 How do you know that the tracking of the sun is sufficiently accurate? (I see some discussion of this point later on).

=> as for manual observations, we have visually controlled the illumination of the quartz plate during the development phase. Later on, we developed the elevation / azimuth scanning routines (§3.4, fig 7) to check remotely the sun pointing in case of doubtful data.

(Figure 4) The CD is more variable from point-to-point and interestingly, also has larger rises and falls than the AD and AC during the short-term variations. Would you like to comment on that?

=> The ozone values being related to the R-dial differences (O3AD $\approx$ RA-RD, O3AC $\approx$ RA-RC and O3CD $\approx$ RC-RD), a constant ïĄďR uncertainty imply smaller (larger) point-to-point variations when R curves difference is large (small). Figure 4 shows the lower variations at the beginning, resp. end of the day compared to mid-day variations when the three R-curves get closer.

Lines 11-12 It sounds like a person has to be at the site quite early in the morning to change from Umkehr to Direct Sun mode? This is disappointing from a pragmatic viewpoint because it reduces some of the benefit of automation.

=> In the discussion, it is mentioned that adding a robotic arm to remove the sun director would allow switching from zenith (Umkehr) to direct sun observations. This extension of the system was not considered as essential at Arosa since we have dedicated Dobson for direct sun and Umkehr observation programs. Such developments

can be foreseen in partnership with other institutes.

Lines 10-12 Are you also performing sunset Umkehrs?

=> Yes we do, our measurements program covers daily sunrise and sunset Umkehrs.

Lines 3-11 Are the results of the lamp tests over a long period of time able to be presented by the software in a user-friendly way? This would be extremely helpful, eg if you could track the lamp tests over a year or in between two calibrations.

=> Historically, the data acquisition and data control programs were distinct and it is still the case today. Our QC/QA software plots the lamp test results but this is not part of the data acquisition software.

Lines 25-30 This addresses the question I asked a couple of pages ago about the accuracy of the sun tracking. It sound like the check is not automatically scheduled but set into action by the operator? Are there any criteria for how close it needs to be?

=> Presently, the sun tracking checks are not set in the regular observation schedule. Our goals were not to have week- or month-long unsupervised operation but a flexible remote control of the Dobsons. The modularity of the code allows setting these tests on a regular schedule if required. Our experience shows that the elevation should be set to $\leq 1°$ and azimuth to $\leq 3°$.

(The peak of the curve on the azimuth plot in figure 7 is quite wide). If the instrument levelling is not quite right of course the tracking might be correct at certain times but not others.

=> As for other instruments (e.g. Brewer, Pandora, etc.) the levelling and the sun pointing is an essential part of the setting up. In our specific set up, the dome confines the sun director in a very limited space requiring a precise levelling.

Line 3 G.M.B. Dobson not G.W. I don't hesitate to say I believe Dobson would consider your S-curves to be very "nice".

=> thanks for the comment

Line 4 It's not clear how automating the "old" instruments extends their lifetimes? Do you mean just that they will be cheaper to operate, or something more?

=> In the present days, the resource and budget reductions as well as the "whole digital" trend would condemned the use of the manually operated Dobson.

Line 7 It's not clear how automating the instruments improves the measurement "quality". Especially in the "discussion" section you should expand on this point.

=> The option of submitting two papers justifies not repeating the arguments in both publications. The present one is oriented on the technical aspects and the other on the data quality analysis.

Lines 9-10 Similarly, you need to explain how automating the instruments will help address these longstanding issues.

=> We added the recent paper from Gröbner et al. 2021 which precisely treats these issues and the advanced processing algorithm.

Line 6 Correct the spelling of "Komhyr"

=> corrected

Lines 13 "his" should be "its"

=> corrected

Lines 4-5 This sentence is not very clear. Isn't the point the high-temporal-resolution, not the +/- 1 DU? Is it useful to be able to follow these small-scale variations?

=> sentence adapted. The traditional used of Dobson for long-term monthly averaged data do not asked for small-scale variations. However, improved resolution (time and quality) for satellites and model validations or for transient phenomenon affecting ozone are example of application of high-resolution measurements.

Lines 5-7 I think you could make it clearer to the reader why the housekeeping data would facilitate further understanding as you claim.

=> the sentence is reformulated

One question many readers would be interested in is what are the advantages and disadvantages of the automated Dobson system compared to a Brewer? (You might consider this to be out of scope for the current work, however).

=> Yes indeed, the scope of this publication is not a Dobson – Brewer comparison even though both instruments types produce very consistent data as developed in Gröbner et al.

Line 2 Albrecht is a strange choice – it's more about the policy issues of the Montreal Protocol

=> it is intentional to sometimes open the perspective showing that reaching an international agreement is not an easy task.

Line 5 I think you could do better here. The link only goes to the EURAMET page.

=> Project Number added

Line 1 Did you mean to refer to a particular chapter or chapters of the 2018 Ozone

Assessment?

=> the reference to Chapter 3 is added

---

## Author Comment (AC2) · 13 Apr 2021

Reply to RC2 comments, AMT-2020-391 manuscript, "A fully Automated Dobson Sun Spectrophotometer for total column ozone and Umkehr measurements" by René Stübi et al."

The authors thank referee#2 for the critical reading and the valuable comments. Also, the numerous suggestions were very helpful to improve our manuscript.

Initial Comments: This submitted manuscript would fit in the category of a commentary, as it describes the modernization of an existing measurement program. I recommend

publication after the issues below are addressed.

Specific Comments/Questions:

I am not commenting on the details of the automation and electronics, as I am not currently experienced in this field.

Introduction:

A reference to the history of the discovery of ozone depletion by chlorofluorocarbons (CFCs) is expected. The discovery of the Antarctic ozone hole by Dr. Solomon is a good source for the history. Suggest a lead-in sentence similar to: The history of the detection of the ozone layer depletion is one of the most important scientific stories of the 20th century (Solomon, 2019).

=> this is a good suggestion that we have adopted

Line 20: The term "calibrate" is not correct. This paragraph should be re-written. The various ground-based and space-based networks have independent calibration methods. Data results from the various instruments and networks are inspected and inter-compared to detect problems within networks. I believe that authors are also saying that the algorithms used to convert data to total ozone values are evolving with increased understanding of the instrument characteristics, and the assumptions used in the measurement and data reduction algorithms.

=> We agree that the term calibrate is not adequate. The paragraph has been re-written

Line 25: There should be some mention of the development of the instrument for the early 1900s' studies in Stratospheric Circulation.

=> The early circulation studies have been added

2 Dobson measurement principle and instrument design:

Table 1: Why is there no A-pair wavelength values for D 062 or FWHM values for D 062 or D 051 ?

=> The intensity of TUpS light source were too low to measure correctly the A-pair of D062. The FWHM of the D062 and D051 are similar to those of D101. A remark is added in the table header.

3.4 Automation of instrument tests

There are other processes in the operation of a Dobson observing program. One of which is determining the attenuation curve of the optical attenuator. Has there been an attempt to automate this process?

=> the calibration of the wedges is carried out as part of the regular intercomparison organized by the regional Dobson calibration center (Hohenpeissenberg for EU). There was no attempt to develop an automated wedge calibration system at MeteoSwiss.

Page 14, line 10: Larger changes are normally a sign of either an aging lamp or a change in the instrument response and are corrected by an update of the attenuator calibration curve. A better explanation is required. The data reduction algorithm incorporates the changes in the standard lamp test values from the lamp values determined at the time of the instrument's calibration by comparison to a reference instrument. The attenuator calibration curve is determined by a different procedure. The standard lamps are actually reference lamps, with measured values for a certain Dobson instrument on a specific date. The change in the measured values with time indicates aging of the instrument. Use of multiple lamps on varying time schedules allows for detection of aging lamps.

=> A rephrasing of the sentence were done

Technical Corrections/Comments/Suggestions

Abstract:

Line 3: Suggest: However, the Dobson sun spectrophotometer requires manual operation which has led to the discontinuation of its use at many stations, thus disrupting long term records of observation.

=> suggestion adopted

Line 8, Suggest: Compared to manual operation, the automation results in a higher number of daily measurements with lower random error and additional housekeeping information to understand the measuring conditions.

=> suggestion adopted

Introduction:

Line 9: Suggest: Moreover, the uncertainties associated with climate change feedback on the ozone recovery process require dedicated ground-based measurement networks for sustained monitoring .

=> suggestion adopted

Line 14: Suggest: The principle of the instrument developed by G. M. B. Dobson in the early 1920s is based on measurements of the intensity of ozone-attenuated radiation in a number of narrow spectral bands. This was first done by analyzing spectra recorded on photographic plates, later directly on spectra within the instrument with photoelectric detectors and nowadays with photo-multipliers (PM) detectors.

=> suggestion adopted

Page 2, Line 32: Suggest: After the discovery of ozone layer depletion by CFCs, the measurement program continued as part of the global effort to verify that the Montreal Protocol was working. The more recent automation of the Dobson operation allows for continuation and improvement of this observation program, with reduced operational cost.

=> the suggestion is included and the paragraph rearranged.

3 Automation of the LKO Dobson instruments

Suggest starting with a sentence similar to: The instrument and observation facilities have had numerous improvements over the years.

=> suggestion is included and the paragraph rearranged.

3.1 Instrument Control

Page 7, Line 7: the local horizon

=> corrected

Page 7, Line 11: Suggest: The sun's image must fall on the entrance window of the instrument, thus the sun's azimuth and elevation must be tracked.

=> suggestion adopted

3.3 Measurements results Page 12, line 12: suggest: has not yet been automated .

=> suggestion adopted

3.4 Automation of instrument tests

Page 14, Line 2, Suggest: Once the measurement procedure had been developed, the data acquisition (DAQ) system could then be programmed to perform other specific tasks .

=> suggestion adopted

Page 14, Line 12: Suggest: A Hg lamp is used to verify the wavelength settings and to check the optical alignment of the Dobson instrument.

=> suggestion adopted

Discussion Page 18. Line 6 : Komhyr not Koomhyr

=> corrected

Page 18, Line 10: Suggest: The Swiss automation system is unique in that the instruments...

=> suggestion adopted

References:

Page 20, Line 20 Komhyr, not Komyhr – references in text (Page2, line 19; Page 4, Line 2) have to be corrected

=> corrected

―――――――――――――――――――